# Metabolic Syndrome and Coronary Artery Disease Risk: A Meta-Analysis of Observational Studies

**DOI:** 10.3390/ijerph18041773

**Published:** 2021-02-11

**Authors:** Amal F. Alshammary, Khalid Khalaf Alharbi, Naif Jameel Alshehri, Vishal Vennu, Imran Ali Khan

**Affiliations:** 1Department of Clinical Laboratory Sciences, College of Applied Medical Sciences, King Saud University, Riyadh 11433, Saudi Arabia; aalshammary@ksu.edu.sa (A.F.A.); kharbi@ksu.edu.sa (K.K.A.); 439106146@student.ksu.edu.sa (N.J.A.); 2Department of Rehabilitation Sciences, College of Applied Medical Sciences, King Saud University, Riyadh 11433, Saudi Arabia; vvennu@ksu.edu.sa

**Keywords:** metabolic syndrome, coronary artery disease, MetS, cardiovascular disease (CVD), CAD

## Abstract

Although numerous studies have described the link between metabolic syndrome (MetS) and Coronary Artery Disease (CAD), no meta-analysis has been carried out on this relationship. Thus, the present study intended to address this limitation. A systematic search was carried out using electronic databases, such as PubMed, CINAHL Plus, Medline, and Web of Science. A sum of 10 studies (*n* = 9327) was incorporated in the meta-analysis. Compared with non-MetS, MetS was significantly associated with high CAD risk (OR = 4.03, 95% CI = 3.56–4.56). The MetS components were also significantly correlated with high CAD risk (OR = 3.72, 95% CI = 3.22–4.40). The presence of two (OR = 3.93, 95% CI = 2.81–5.49), three (OR = 4.09, 95% CI = 2.85–5.86), four (OR = 4.04, 95% CI = 2.83–5.78), or all five MetS components (OR = 3.92, 95% CI = 3.11–4.93), were significantly associated with a high risk of CAD. MetS and its individual or combined elements were linked with high CAD risk based on contemporary evidence. Thus, the assessment of MetS and its components might help identify people at a higher risk of advancing CAD in the future.

## 1. Introduction

The World Health Organization (WHO), the European Group for the study of insulin resistance, the 2005 National Cholesterol Education Program (NCEP)-Adult Treatment Panel III (ATP-III), and the 2006 International Diabetes Federation formulated the definition of Metabolic Syndrome (MetS) after great debate [1,2]. MetS is a principal public health burden worldwide, and involves five cardiovascular hazard circumstances: obesity or high waist circumference (WC), high blood pressure (BP), hyperglycemia, hypertriglyceridemia, and low high-density lipoprotein cholesterol (HDL-c) levels [3,4]. MetS and its five individual components affect the blood vessels. As a result it is associated with extending the risk of cardiovascular disease by two-fold (CVD), type 2 diabetes mellitus (T2DM ) by five-fold [5], and all-cause mortality by 1.5 fold [6,7], along with elevating some specific cancer risks [8,9]. In recent years, the ubiquity and frequency of MetS have quickly progressed globally [10].

The enhanced pervasiveness and degree of MetS and its elements are causing harmful effects on coronary circulation [11]. However, the primary pathways connecting MetS and coronary artery disease (CAD) remain complex [12]. The connection between MetS and CVD in diverse communities is indicated [13,14,15]. However, no meta-analysis has been accompanied to examine the relationship between MetS and CAD risk. There is an urgent need to investigate this relationship by taking public health into account due to the high rate of MetS globally and the impact of CAD on coronary arteries the heart muscles. 

While differently accepted definitions have been used in the published observational studies, the current meta-analysis of these studies tested the association of MetS and its segments with CAD risk. The primary hypothesis is that MetS would associate significantly with CAD risk. Furthermore, MetS components both individually and in combination would significantly associate with CAD risk.

## 2. Materials and Methods

The current meta-analysis was conducted based on a recommendation by the Meta-analysis of Observational Studies in Epidemiology (MOOSE) group [16]. Literature exploration, selection criteria, data extraction, and statistical analysis were carried out based on the Cochrane Collaboration guidelines [17].

### 2.1. Search Strategy

As advised by MOOSE [16], the search was carried using electronic databases, such as PubMed, CINAHL Plus, Medline, and Web of Science. These keywords or MeSH terms were utilized for the database search: MetS, CAD. The search was carried out for observational studies published till 25 September 2020. The search was also restricted to studies including adults aged 19 years or above and published as full-text in English.

### 2.2. Inclusion and Exclusion Criteria

The studies involved met the following inclusion criteria. The observational studies used for this work included MetS and its components as exposure and CAD as an outcome in adults aged 19 years and more. MetS components were as follows: (1) obesity or high WC, (2) advanced BP, (3) low HDL-c, (4) raised triglycerides (TG), and (5) raised hyperglycemia. MetS and its composition has been defined elsewhere [18]. 

Patients who have not experienced a recent acute incident of a known cardiovascular disease are defined as CAD [19]. CAD is also defined as restricted bloodstream to the heart muscle upon the build-up of fats, cholesterol, and plaque (atherosclerosis) in the heart’s arteries [20]. Clinically, CAD is determined as the presence of any coronary artery calcium score > 100, >75th percentile for one’s age and sex, or > 400 [21]. The development of CAD is determined according to the coronary artery stenosis as follows: (1) normal (no plaque and no stenosis to a plaque with < 25% stenosis); (2) mild (25%–49% stenosis); (3) moderate (50%–69% stenosis); (4) severe (70%–99% stenosis); and (5) occluded (100% stenosis) [22,23]. Table 1 presents the different accepted definitions of MetS and CAD in the studies incorporated in the meta-analysis.

### 2.3. Data Extraction

Data were individually filtered by extracting potentially relevant studies based on the title and abstract. According to the eligibility criteria, the relevant studies were selected separately by reading the full-text. The final eligible studies were then chosen according to the inclusion and exclusion standards.

In this meta-analysis, we included prior studies that reported odds ratios (ORs) and 95% confidence interval (CI) for the association of MetS and its segments to CAD risk compared to non-MetS. The exclusion criteria, OR and CI participants with a diagnosis other than MetS and its components were not reported. Studies that did not explore the link between MetS, its components, and CAD risk were eliminated.

### 2.4. Statistical Analysis

The primary outcome was the OR of CAD in people with MetS compared to non-MetS subjects. The secondary outcome was the OR of CAD in individuals with any element of MetS. The territory outcome was the OR of CAD in individuals with two or more MetS segments. A satisfactory agreement was defined as adjusting at least two of six covariates, including age, sex, family records of CAD, body mass index (BMI), low-density cholesterol, and smoking status. 

We scrutinized multivariate-adjusted outcome data, displayed as OR and 95% CI. The reversed variation method was applied for investigation after log-transformation of study-specific assessments. For example, 95% CI was converted into standard error (SE) employing the following equation: SE = upper CI - lower CI/3.92. The I^2^ statistic was applied to test heterogeneity. The heterogeneity between studies was interpreted as moderate (30%–59%), substantial (60%–89%), or considerable (90%–100%) [17]. Studies were combined, applying a fixed-effect model, if there was a relevant, later state of heterogeneity among studies. Otherwise, a random-effects model was used. The potential publication bias for primary and secondary outcomes was assessed utilizing funnel plots [32]. All analyses were completed using RevMan software version 5.4 for Windows (The Cochrane Collaboration, Copenhagen, Denmark) [33]. The *p* < 0.05 was described as of statistical importance. 

## 3. Results

### 3.1. Selected Studies and Characteristics

Out of 1268 studies identified in a database search, a total of 10 studies [22,23,24,25,26,27,28,29,30,31] were incorporated in the meta-analysis after eliminating those not falling within the eligibility criteria. Excluded articles and the reason for their exclusion are illustrated in Figure 1. The features of the incorporated studies are described in Table 2. A total of 9327 subjects participated in the ten included studies. Four studies were from South Korea [23,26,27,28], and three studies were from the United States of America (USA) [22,24,30]. Of 9327 subjects, 3969 had MetS and are referred to as cases, while 5358 had no MetS and are referred to as controls. Out of 3969 cases, 1948 were men, and 2021 were women. Men, *n* = 3048 and women, *n* = 2310 were controls. The average age of all participants was above 45 years.

### 3.2. Association between MetS and CAD

Figure 2 shows the link between MetS and CAD. Nine studies [22,23,24,25,26,27,28,30,31], with 8879 participants reported CAD ORs with MetS groups compared with non-MetS groups. The fixed-effect pattern was applied to analyze the correlation between MetS and CAD. There was no notable heterogeneity between all studies (I^2^ = 13%, *p* = 0.33). MetS was significantly associated with high CAD risk (OR = 4.03, 95% CI = 3.56–4.56).

### 3.3. Association between Any MetS Component and CAD

Figure 3 exhibits the association between different MetS components and CAD risk. Five studies [24,25,26,28,30] with 4527 participants reported CAD ORs with high fasting blood glucose (FBG) groups compared to low FBG groups. The random-effect design was employed to analyze the connection between high FBG and CAD risk due to substantial heterogeneity between studies (I^2^ = 67%, *p* = 0.02). 

Six studies [22,24,25,26,28,30] with 4389 participants reported ORs of CAD with low HDL-c, high TG, and high BMI groups compared with high HDL-c, low TG, and low BMI groups. Significant heterogeneity was found in low HDL-C (I^2^ = 79%, *p* = 0.003), high TG (I^2^ = 66%, *p* = 0.01), and high BMI (I^2^ = 60%, *p* = 0.03) groups, within the six studies. 

Five studies [22,25,26,28,30] with 3603 participants reported ORs of CAD with a high BP group compared with the low BP group. Notable heterogeneity was found between all five studies (I^2^ = 83%, *p* = 0.01). Significant heterogeneity was found among all studies that reported ORs of CAD with any component of MetS (I^2^ = 76%, *p* = 0.01). Thus, any MetS component was correlated significantly with high CAD risk (OR = 3.72, 95% CI = 3.22–4.40).

### 3.4. Association between Multiple MetS Components and CAD

Figure 4 represents the association between two or more MetS elements and CAD risk. Seven studies [22,24,25,26,28,30,31] with 5598 participants reported CAD ORs with two MetS component groups compared to non-MetS groups. There was important heterogeneity between all seven studies (I2 = 76%, *p* = 0.004). Thus, two MetS components’ presence was significantly associated with high CAD risk (OR = 3.93, 95% CI = 2.81–5.49).

Six studies [22,24,25,26,28,30] with 4389 participants reported CAD ORs with three MetS components compared with non-MetS component. Significant heterogeneity was found between all six studies (I2 = 75%, *p* = 0.001). The presence of three MetS components was significantly associated with high CAD risk (OR = 4.09, 95% CI = 2.85–5.86). Six studies [22,24,25,26,28,30] with 4389 participants reported CAD ORs with four MetS components compared with non-MetS component. Significant heterogeneity was found between all six studies (I2 = 71%, *p* = 0.04). The presence of four MetS components was significantly associated with high CAD risk (OR = 4.04, 95% CI = 2.83–5.78).

Four studies [25,26,28,30] with 3741 participants reported CAD ORs with all five MetS components compared with non-MetS components. The fixed-model was adopted for the analysis due to no significant heterogeneity between all four studies (I2 = 27%, *p* = 0.25). The presence of all five MetS components was significantly associated with high CAD risk (OR = 3.92, 95% CI = 3.11–4.93).

### 3.5. Funnel Plot Analysis

There is no bias between the studies in the funnel plot associated with MetS, any of the five MetS, and CAD (Figure 5). The association of two, three or four MetS components and CAD reveals an asymmetric scatter plot (Figure 6 and Figure 7A–C). Figure 7D illustrate the association between all five MetS elements and CAD. 

## 4. Discussion

The current meta-analysis examined the association of MetS and its five elements with CAD risk among adult populations. The findings revealed that MetS correlated significantly with high CAD risk compared to non-MetS. Furthermore, any individual MetS element was significantly linked with high CAD risk. The attendance of two or more MetS components was also connected with high CAD risk compared to the absence of any component. The presence of from two to four MetS components significantly increased the correlation in the severity of CAD. To our understanding, this is the first meta-analysis that presented evidence on the association of MetS and its elements with CAD risk. 

A longitudinal study concentrated on the consequences of MetS on BP control and its function in stimulation of arterial aging in hypertensive treated people [34]. The results revealed that the presence of MetS presents a worse BP control than non-MetS and is correlated with accelerated aortic stiffening. These findings may be partially linked to change in BP. However, this finding must be interpreted with caution due to the lack of BP measures other than supine BP when the pulse wave velocity evaluation is 24 h-Ambulatory BP.

MetS is likely to result in the non-alcoholic fatty liver condition, T2DM, and vascular conditions such as stroke, peripheral vascular disease, and CVD [35]. Initial meta-analysis showed some limited evidence that MetS is a significant risk factor for CVD [36,37]. Another meta-analysis showed that MetS might be a significant hazard factor for stroke [38]. However, to date, no meta-analysis was carried out to evaluate the association of MetS and its elements with CAD risk. Therefore, the current meta-analysis was conducted with every published study to date.

There were no randomized control trials (RCTs) observed during the literature search on this subject. The majority of the published retrospective studies fitted the inclusion standards added to this meta-analysis [22,25,28,31]. There were only two published case-control studies [24,30] that were included. A careful review of all the incorporated studies was conducted for any potential bias.

The relation between MetS and CAD risk has been noted in individual studies among various populations aged 50 years or above. These studies vary considerably in their methodology and MetS definition. Various stratified analyses were conducted to examine possible heterogeneity causes, and the outcomes helped our overall conclusions. The sign of a relationship between MetS and CAD risk appeared with no significant heterogeneity between the studies.

Our findings are comparable with preliminary proof regarding MetS elements. Waist circumference [39], elevated BP [40,41], low HDL-c levels [42,43,44], elevated triglycerides [45,46], and high hyperglycemia [47,48] are individually associated with CAD risk. A study also exhibited a significant connection between one component of MetS and CAD risk development compared two components of MetS in older people [49]. However, all these studies differ from our investigation concerning gender participation and age. A study finding demonstrates that MetS and high BP are linked with enhanced carotid atherosclerosis more than MetS without high BP. The determination of MetS per se might not sufficiently identify subjects at raised cardiovascular risk [50]. Therefore, it might be essential to examine the relationship between MetS elements and CAD risk.

In the meta-analysis, as mentioned earlier, the association of two or more MetS components with CAD risk was similar to an earlier study [51]. That confirmed the association of low levels of HDL-c and high levels of TG to CAD risk. Another study indicated that abdominal obesity and related atherogenic dyslipidemia modulate CAD risk related to hyperglycemia [52]. However, previous study results revealed a significant rise in MetS elements, such as low-HDL-c, high-FBG, and WC, in patients with CAD [37]. This study concluded that hyperglycemia, dyslipidemia, and obesity are indispensable predictors of MetS in patients with CAD.

This meta-analysis’ major strength included studies that defined MetS well, along with its elements (such as WC, BP, HDL-c, TG, and hyperglycemia), and CAD. Another strength of this investigation is the large sample size of 9327 participants from 10 incorporated studies. However, some limitations have been acknowledged when understanding the results of this meta-analysis. First, there were no prospective studies on this topic. This kind of evidence may give more precise evaluations than retrospective studies, because they usually have more reliable controls for possible confounders, including socioeconomic, lifestyle, and health factors related to MetS progress. They also have adequate continuance of follow-up to identify MetS cases.

Moreover, these studies collect susceptibility information before the appearance of diseases. Therefore, the temporal causal relationship can be examined. Secondly, subgroup analyses, such as sex (men versus women) and follow-up duration, were not performed due to the lack of data concerning sex and variation in the study design. Thirdly, inter-study variations in MetS definition and patients’ demographics might also influence the results. Fourthly, the studies that did not show output in OR and 95% CI were excluded. This might decrease the validity of this study. Overall, this is an observational meta-analysis study showing the association, but not the interconnection significance.

The current findings are significant for public health since the ubiquity and frequency of MetS have rapidly increased globally [10]. A possible reason for the increase of MetS incidence might be the western lifestyle, marked by extensive consumption of red and treated meat [53], processed grains and deep-fried foods [54], and sugar-sweetened drinks, such as soft drinks, fruit drinks, iced tea, and energy and vitamin water drinks [55]. Another possible reason might be that many people worldwide do not meet their dairy requirements, especially in the developing countries [56,57], and lack physical activity [54]. The results from this meta-analysis with recent evidence on T2DM [5], all-cause mortality [6,7], and specific cancer risks [8,9] provide further support for public health advice on increasing the consumption of dairy products [58], fruits/vegetables [59], and increasing physical activity [54], limiting the risk of MetS, its components, and CVD, especially CAD, in the global population.

The relationship between MetS, all five components of MetS, and CAD indicates a symmetrical scatter pattern with no publication bias. A potential reaction could be a fixed-effect analysis that assumes little or low heterogeneity and a single underlying effect across studies [32]. Asymmetrical scatter plots of the association between two, three or four MetS components and CAD indicate a publication bias between the studies. this may be due to the studies’ quality, the different intensity of the intervention, and the underlying risk [32]. In addition, random-effect analysis allows for variability and average treatment effect across studies. Although the Funnel plots’ purpose is to detect publication bias, the components of MetS relating to CAD risk, demonstrating an asymmetry of high FBG and low HDL-c, suggests that these components might pose higher CAD risk than high TG and high BP.

## 5. Conclusions

In conclusion, the present meta-analysis intended to examine the association of MetS and its five elements with CAD risk. This meta-analysis shows that MetS and its elements are significantly associated with high CAD risk. The initial evaluation of MetS and its five components might help identify individuals at higher risk of CAD. The current findings suggest that MetS and its components can be tested to validate CAD risk. Future study is required for establishing high-quality evidence by including RCTs or prospective cohort studies in the identification of the link between MetS and CAD risk among the global population.

## Figures and Tables

**Figure 1 ijerph-18-01773-f001:**
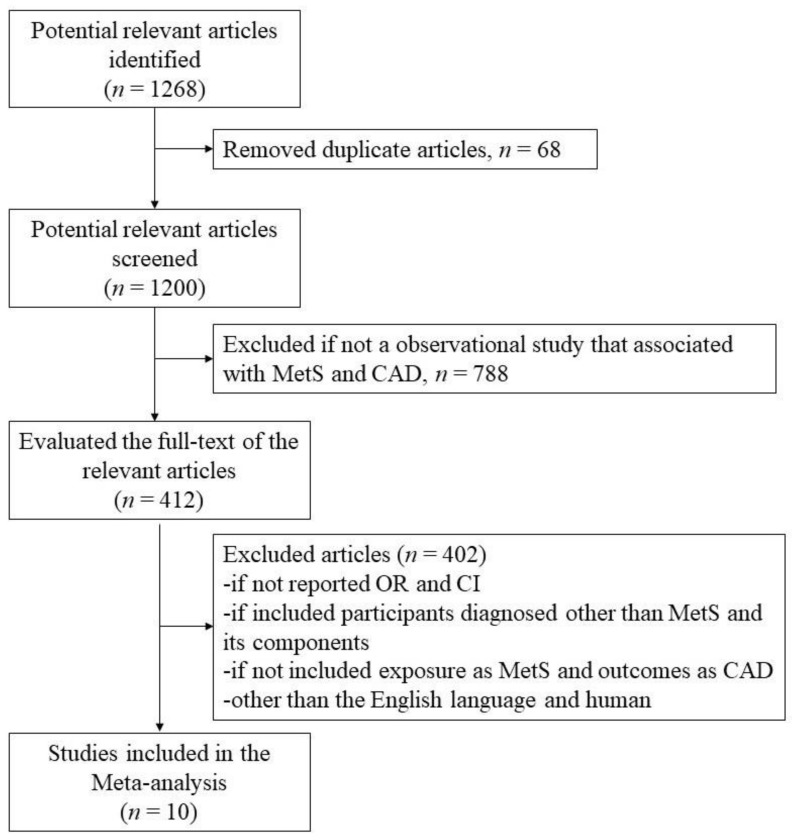
Flowchart of studies identified through a database search.

**Figure 2 ijerph-18-01773-f002:**
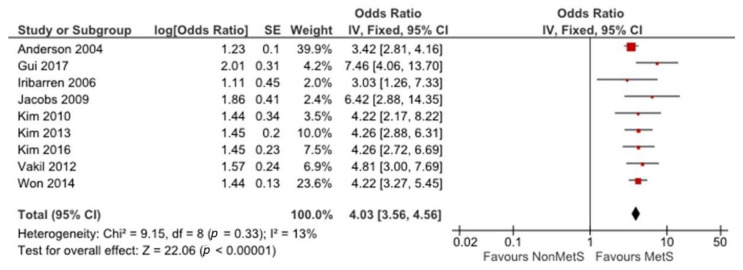
The association between MetS and CAD risk. The red symbols indicate individual studies’ odds ratios and black line shows 95% CI. The horizontal points of the diamond show the studies average limits of the 95% CI.

**Figure 3 ijerph-18-01773-f003:**
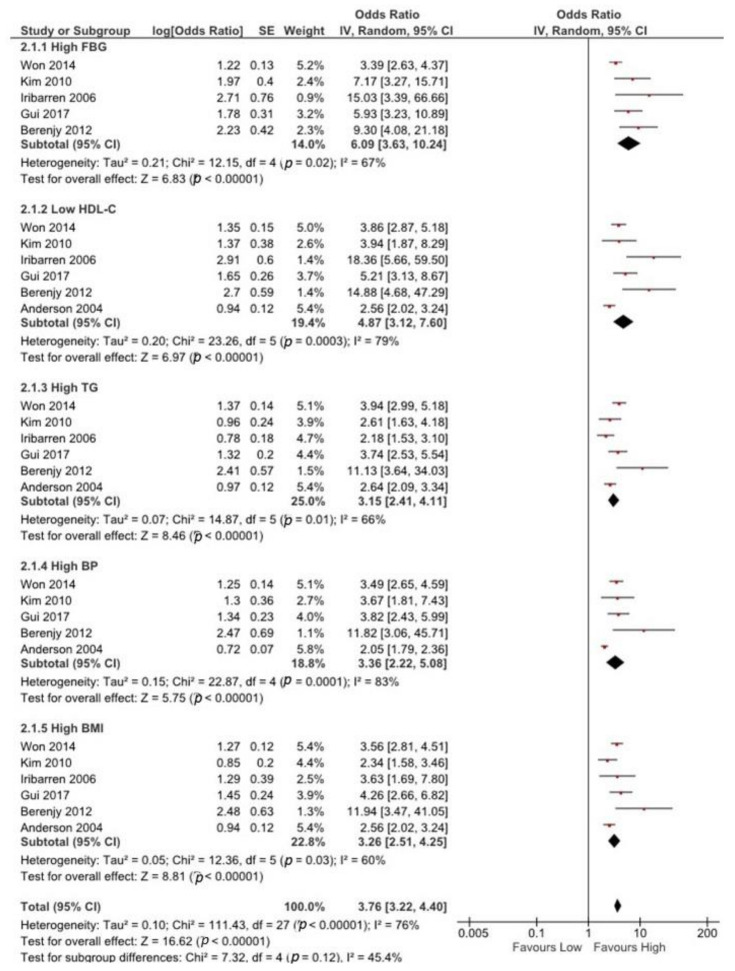
The association between any MetS component and CAD risk. The red symbols indicate individual studies’ odds ratios and black line shows 95% CI. The horizontal points of the diamond show the studies average limits of the 95% CI.

**Figure 4 ijerph-18-01773-f004:**
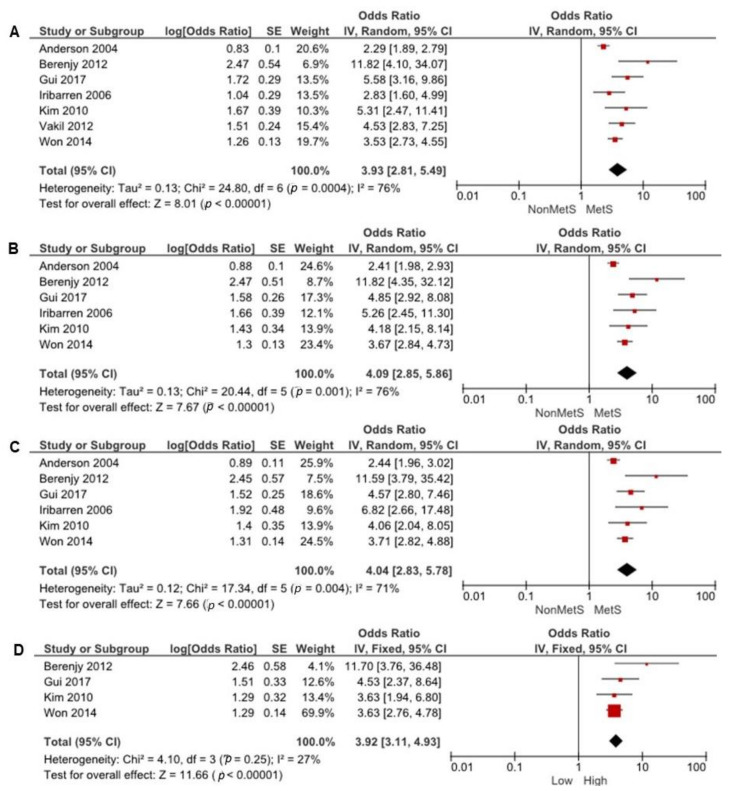
The association of (**A**) two, (**B**) three, (**C**) four, and (**D**) all five MetS elements with CAD risk. The red symbols indicate individual studies’ odds ratios and black line shows 95% CI. The horizontal points of the diamond show the studies average limits of the 95% CI.

**Figure 5 ijerph-18-01773-f005:**
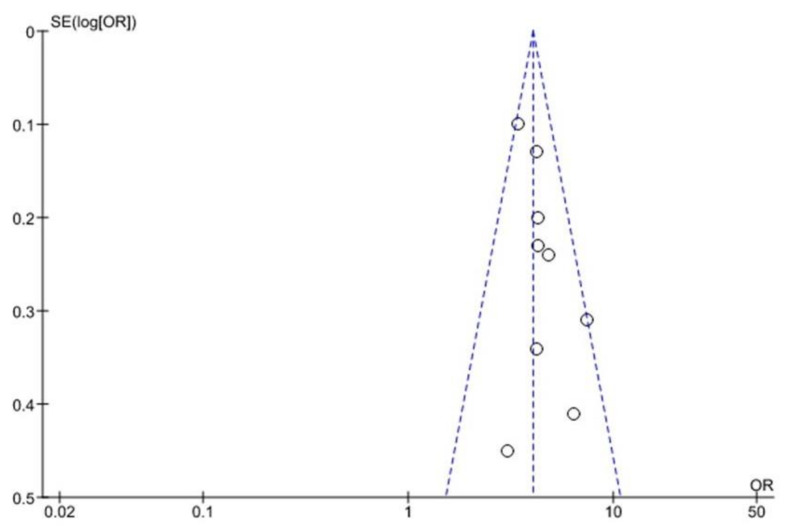
Association between MetS and CAD outcome. The dashed lines indicate the funnel and open circles represents effect from individual studies.

**Figure 6 ijerph-18-01773-f006:**
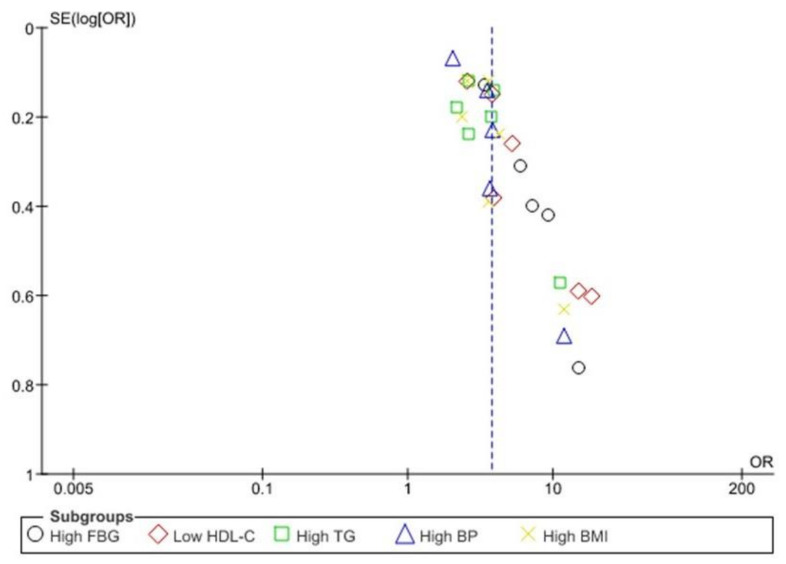
Association between MetS individual components and CAD outcome.

**Figure 7 ijerph-18-01773-f007:**
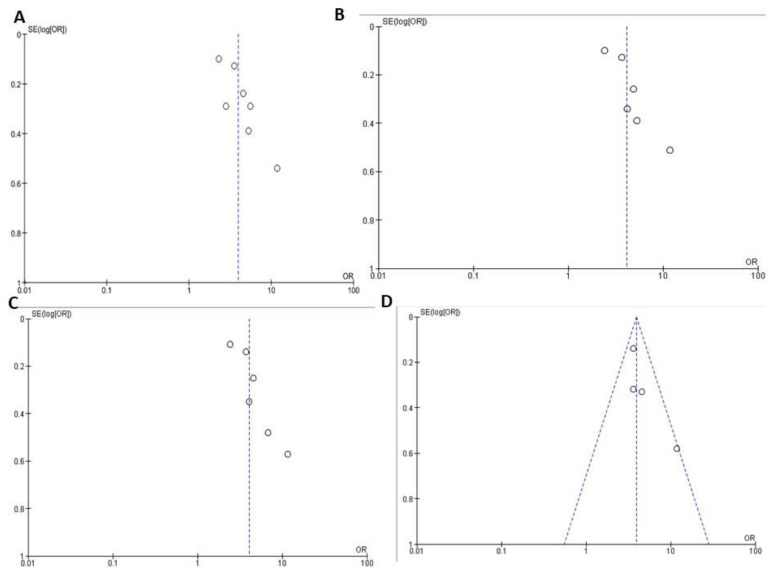
Association between the presence of (**A**) 2, (**B**) 3, (**C**) 4 and (**D**) all 5 MetS components and CAD outcome. The dashed lines indicate the funnel and open circles represents effect from individual studies.

**Table 1 ijerph-18-01773-t001:** MetS and CAD definitions as per the studies incorporated in this meta-analysis.

Study	MetS	CAD
Iribarren et al. [24]	The 2001 NCEP-ATP-III definition (1) (WC ≥102 cm in men or ≥88 cm in women, fasting TG ≥150 mg/dL, HDL-c <40 mg/dl in men or <50 mg/dL in women, BP ≥130/85 mm Hg or use of BP medication, and FBG ≥110 mg/dL). The revised American Heart Association/National Heart, Lung, and Blood Institute definition incorporating the lower threshold for impaired fasting glucose (≥100 mg/dL) and FBG ≥126 mg/dL	The ninth revision of the International Classification of Disease codes 410.x, 411.x, 413.x, or 414.x plus troponin. I > 4ng/dL, combination of creatine kinase fraction% >3.3, symptoms plus an electrocardiogram, and evidence of angiography during hospitalization.
Gui et al. [25]	The presence of ≥3 of the following five criteria: (1) WC: ≥90 cm in men or ≥80 cm in women; (2) TGs: ≥1.7 mmol/L or on drug treatment for elevated TGs; (3) HDL-c: <1.03 mmol/L in men or <1.3 mmol/L in women or on drug treatment for HDL-c; (4) BP ≥130 mm Hg, ≥85 mm Hg, or on antihypertensive drug treatment; (5) FBG ≥5.6 mmol/L or on drug treatment for elevated glucose.	Fifty percent or more lumen diameter reduction in at least one major coronary artery.
Won et al. [26]	The presence of ≥3 of the following: (a) BP ≥130 mm Hg, ≥85 mm Hg diastolic, or on antihypertensive treatment; (b) HDL-c <40 mg/dL in men or <50 mg/dL in women; (c) TG ≥150 mg/dL; (d) BMI ≥25 kg/m2; and (e) FBG ≥100 mg/dL.	At least one of the symptoms, including typical angina, atypical angina, dyspnea, and excessive fatigue, but not an acute coronary syndrome, required emergent coronary intervention or surgery.
Kim et al. [27]	The presence of MetS was determined using the updated 2005 ATP-III of the NCEP criteria by considering WC ≥90 cm in men and ≥80 cm in women. Those who had ≥3 of the five components were classified as having MetS.	First elective coronary angiography for evaluation of chest pain.
Kim et al. [23]	The presence of ≥3 of the flowing five criteria: (1) WC ≥90 cm in men and ≥80 cm in women, (2) TG ≥150 mg/ dL (1.7 mmol/L); (3) HDL-c <40 mg/dL (1.0 mmol/L) in men and <50 mg/dL (1.3 mmol/L) in women; (4) BP ≥130/85 mm Hg or the use of antihypertensive medication; and (5) FBG ≥100 mg/dL (6.1 mmol/L) or the self-reported use of antidiabetic medication (insulin or oral agents).	CAD assessed by the degree of coronary artery stenosis, presence of plaques, and its characteristics.
Kim et al. [28]	The presence of ≥3 of the following: WC modified into a body mass index of more than 25 kg/m^2^. The BP >130/85 mm Hg or being already on antihypertensive medication. FBG ≥110 mg/dL or being already on oral hypoglycemic agents or insulin. HDL-c < 40 mg/dL in men and < 50 mg/dL in women. TG ≥150 mg/dL.	>50% luminal diameter stenosis of at least one major epicardial coronary artery.
Jacobs et al. [29]	According to the American Heart Association and the National Heart, Lung, and Blood Institute, it is defined.	Self-reported myocardial infarction, bypass surgery of the coronary arteries, balloon dilatation or stent placement, and/or the presence of signs of myocardial infarction (Minnesota codes 1–1 or 1–2) or ischemia (Minnesota codes 1–3, 4–1, 4–2, 4–3, 5–1, 5–2, 5–3 or 7–1) on a 12-lead electrocardiogram.
Anderson et al. [22]	A combination of ≥3 of the following features: FBG 110 mg/dL or more, TG 150 mg/dL or more, HDL-c <40 mg/dL in men or <50 mg/dL in women, BP 130/85 mm Hg or more and WC >40 inches in men or more than 35 inches in women.	The degree of maximal diameter stenosis of each of the three principal coronary arteries and their major branches was estimated to the nearest 10%.
Berenjy et al. [30]	WC >88 cm in women and >102 in men. TG ≥150 mg/dL or medication for treatment. HDL-c <50 mg/dL in men and <40 in women or medication for treatment. BP ≥ 130/85 mm Hg or medication for treatment. FBGF ≥ 100 mg/dL or medication for treatment.	>50% luminal diameter stenosis of at least one major epicardial coronary artery.
Vakil et al. [31]	As per the revised NCEP, the presence of ≥3 of the following: A WC >102 cm, BP ≥ 130/85 mm Hg or on treatment, FBG ≥ 100 mg/dL, HDL-c ≥ 40 mg/dL, TG ≥ 150 mg/dL. As per the International Diabetes Federation, the presence of a WC ≥ 94 cm and at least two following risk factors: BP ≥ 130/85 mm Hg or on treatment, FBG ≥ 100 mg/dL, HDL-c < 40 mg/dL, TG ≥ 150 mg/dL.	The presence of reversible and/or irreversible perfusion defects on single-photon emission computed tomography or resting and/or stress-induced wall motion abnormalities on two-dimensional stress echocardiography.

**Table 2 ijerph-18-01773-t002:** Descriptive characteristics of included studies.

Author	Study Design	Country	Sample Size	Age in Year Mean (Standard Deviation)	Exposure	Risk Factors Adjusted	Outcome
Irribarren et al., [24]	Case-control	USA	*n* = 393 (women = 239) cases *n* = 393 (women = 239) control	Case: 45.8 (6.5) Control: 45.2 (5.6)	Metabolic Syndrome (MetS)	Age, educational attainment, cigarette smoking, alcohol consumption, and body mass index	Early-onset Coronary Artery Disease (CAD)
Gui et al., [25]	Retrospective	China	*n* = 296 (women = 91) without MetS *n* = 330 (women = 141) with MetS	58 (9) without MetS 60 (9) with MetS	MetS score, MetS, and its components	Age and sex	Angiographic CAD
Won et al., [26]	Cross-sectional	S. Korea	*n* = 1515 (men = 749) without MetS *n* = 793 (men = 406) with MetS	57 (9) with no MetS 57 (9) with MetS	MetS and its components	Age, sex, current smoking, and low-density cholesterol	Severity of CAD
Kim et al., [27]	Cross-sectional	S. Korea	*n* = 185 (male = 100) without MetS *n* = 178 (male = 96) with MetS	61.8 (11.5) without MetS 62.2 (10.6) with MetS	MetS score in the subject without diabetes included	Age and sex	Angiographic CAD along with high sensitivity C-reactive protein, interleukin-6, resistin, and adiponectin
Kim et al., [23]	Retrospective cohort	S. Korea	*n* = 825 (male = 661) with MetS *n* = 1601 (male = 1278) without MetS	56.6 (7.5) with MetS 56.2 (7.2) without MetS	MetS	Age, sex, smoking status, family history of CAD, body mass index, and low-density cholesterol	Progression of CAD
Kim et al., [28]	Retrospective	S. Korea	*n* = 349 (male = 223) without MetS *n* = 283 (male = 171) with MetS	61 (10.8) without MetS 61 (10.4) with MetS	MetS and its components	MetS components	Angiographic CAD
Jacobs et al., [29]	Cohort	The Netherlands	*n* = 225 (male = 149) without MetS *n* = 305 (male = 194) with MetS	58.8 (7.4) without MetS 60.2 (6.6) with MetS	MetS	Age, sex, and smoking	CAD
Anderson et al., [22]	Retrospective	USA	*n* = 69 male without MetS *n* = 69 male with MetS	63 (13) without MetS 63 (12) with MetS	MetS and its components	Age, sex, smoking status, family history of CAD, body mass index, and low-density cholesterol	Angiographic CAD
Berenjy et al., [30]	Case-control	Malaysia	*n* = 258 cases with MetS *n* = 190 control without MetS	-	MetS components	-	CAD
Vakil et al., [31]	Retrospective	USA	*n* = 1071 male with or without silent CAD	61 (11)	MetS and its component, such as waist circumference	-	Silent CAD

## Data Availability

Not applicable.

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
