# Peer review of "Metabolic Syndrome and Coronary Artery Disease Risk: A Meta-Analysis of Observational Studies"

_ijerph, 2021, doi:10.3390/ijerph18041773_

Round 1

Reviewer 1 Report

Thank you for the opportunity to review the paper. Alharbi et al present a meta-analysis of observational studies to study the relationship between metabolic syndrome and coronary artery disease. Metabolic syndrome is a group of medical conditions which has been shown to be associated with increased risk of CAD and also individual components have been shown to have an association with CAD. 

The limitation of this meta-analysis is all observational studies and high heterogeneity in the analysis. However, this is an extremely important public health hazard. I congratulate the authors on the use of sound methadology for meta- analysis.  Please correct few grammatical mistakes and some sentences i.e."Given the high rate of MetS globally and the consequence of CAD on coronary arteries that enhance the heart muscles, there is an urgent need, considering the public health, to examine the association of MetS and its components with CAD's risk" besides others need thorough English language review. 

Author Response

Reviewer 1

Thank you for the opportunity to review the paper. Alharbi et al present a meta-analysis of observational studies to study the relationship between metabolic syndrome and coronary artery disease. Metabolic syndrome is a group of medical conditions which has been shown to be associated with increased risk of CAD and also individual components have been shown to have an association with CAD.

  1. Q) The limitation of this meta-analysis is all observational studies and high heterogeneity in the analysis. However, this is an extremely important public health hazard. I congratulate the authors on the use of sound methodology for meta- analysis. Please correct few grammatical mistakes and some sentences i.e. “Given the high rate of MetS globally and the consequence of CAD on coronary arteries that enhance the heart muscles, there is an urgent need, considering the public health, to examine the association of MetS and its components with CAD's risk" besides others need thorough English language review
  2. A) We are deeply thankful to the reviewer for valuable comments. The manuscript has been edited concerning few grammatical mistakes and the English language review by the author individually.

Reviewer 2 Report

Proposed paper is interesting and well written. However some revisions are needed before it can be accepted for pubblication.

  • OR doesn't increase depending on the number of components. This need to be discussed in the relative section. Is MS pro-atherogenic per se independently on the number of compoents? this could be read in accordance with the hypothesis that MS is a single disease and not an exponential sum of different metabolic disease. Please discuss.
  • Many different definition of MS are present although the most used is the ATP-III one. How this problem has been handled in the metanalysis. Only ATP-III definition were used. I'm not able to find such an information into the paper now.
  • On the same issue also CAD could be defined in very different way. Please explain.
  • Regarding this last two point, if different definition has been accepted and used in the metanalaysis this need to be clearly stated in all the section (introduction, methods, results and discussion) since it represent an important limitation. Furthermore a table with the definition used in all the included paper need to be added.
  • A possible link between MS and CAD could be also arterial stiffness. A recent published work on the topic (Eur J Intern Med. 2020 Jun;76:107-109. ) could help to add some new paragraph to the discussion in order to take into account this possibilities.
  • Many

Author Response

Reviewer 2

  1. Q) Proposed paper is interesting and well written. However, some revisions are needed before it can be accepted for publication.
  2. A) We thank the reviewer for the valuable feedback. We are happy to revise the manuscript with the required and requested amendments in the revised manuscript.

  1. Q) OR doesn't increase depending on the number of components. This need to be discussed in the relative section. Is MS pro-atherogenic per se independently on the number of components? This could be read in accordance with the hypothesis that MS is a single disease and not an exponential sum of different metabolic disease. Please discuss.
  2. A) Thank you for the reviewer for this important comment. As suggested, it has been discussed in the relevant area of discussion section on page 10, lines 235-239.

  1. Q) Many different definitions of MS are present although the most used is the ATP-III one. How this problem has been handled in the metanalysis. Only ATP-III definition were used. I'm not able to find such an information into the paper now.
  2. A) Thank you for the reviewer for this valuable comment. We agree that many different definitions of MetS are present. However, the information has been presented in Table 1.

  1. Q) On the same issue also, CAD could be defined in very different way. Please explain.
  2. A) As suggested, CAD has been defined in very different way on page 4 and 5, lines 104-107.

  1. Q) Regarding this last two point, if different definition has been accepted and used in the metanalaysis this need to be clearly stated in all the section (introduction, methods, results and discussion) since it represents an important limitation. Furthermore, a table with the definition used in all the included paper need to be added.

A possible link between MS and CAD could be also arterial stiffness. A recent published work on the topic (Eur J Intern Med. 2020 Jun; 76:107-109.) could help to add some new paragraph to the discussion in order to consider this possibility. Many

Reviewer 3 Report

General Comment:  The authors conducted a meta-analysis to determine if metabolic syndrome (MetS) and its components associate with CAD.  The analysis appears to be quite thorough and the results are convincing that such an association indeed exists. Overall, the paper is well-written and the conclusions are supported by the data which are well-presented.  I certainly agree that patients diagnosed with MetS present as candidates for CAD.   

Concerns or suggestions:

Do the funnel plots of individual components and 2, 3, or 4 components not require diagonal lines? 

  1. I agree that this study is needed since this specific association has not been reported at least to my knowledge and that of the authors. It would have been quite surprising to me if the results demonstrated no association.  It’s common for clinical organizations to encourage annual screening of adults for MetS based on the risk for CAD and it will be good to have an actual study demonstrating that risk.  Did the authors consider stating a hypothesis in the paper and further breaking that into hypotheses with each component of metabolic syndrome?  
  2. I suggest expanding on the definition and description of CAD. Authors define the components of metabolic syndrome and give a good description of its origin of definition.   The statement on CAD only refers to “…enhance the heart muscles”.  What specifically does this mean – myocyte hypertrophy?  What is the definition of CAD and how is it determined clinically?  Are there degrees of CAD – mild, moderate, and severe?  Does the severity of MetS (e.g., number of MetS components present) correlate with the severity of CAD?    
  3. In the Results, authors state there are four studies from South Korea and three from the U.S. Where do the other three come from? 
  4. I agree that further studies are needed to understand the full significance of the components on MetS on CAD and other pathologies. Is it possible to discriminate among the components of MetS on CAD from the data presented in this study?  Although the purpose of Funnel plots is for publication bias, can one interpret the asymmetry of high FBG in Figure 6 to be suggestive that this component might pose a higher risk for CAD than the high TG or high BP? 
  5. Do the funnel plots of individual components and 2, 3, or 4 components not require diagonal lines?
  6. First paragraph page 13: “These evidence..” - This evidence

Author Response

Reviewer 3

General Comment:  The authors conducted a meta-analysis to determine if metabolic syndrome (MetS) and its components associate with CAD.  The analysis appears to be quite thorough and the results are convincing that such an association indeed exists. Overall, the paper is well-written and the conclusions are supported by the data which are well-presented.  I certainly agree that patients diagnosed with MetS present as candidates for CAD.

  1. A) We are deeply thankful to the reviewer for positive comments on our paper.

Concerns or suggestions:

  1. Q) Do the funnel plots of individual components and 2, 3, or 4 components not require diagonal lines?

  1. I agree that this study is needed since this specific association has not been reported at least to my knowledge and that of the authors. It would have been quite surprising to me if the results demonstrated no association.  It’s common for clinical organizations to encourage annual screening of adults for MetS based on the risk for CAD and it will be good to have an actual study demonstrating that risk. 
  1. Did the authors consider stating a hypothesis in the paper and further breaking that into hypotheses with each component of metabolic syndrome?
  2. Thank you for the reviewer for this critical comment. We agree. As suggested, the hypothesis has been stated by further breaking that into hypotheses on pages 3 and 4, lines 83-85.

  1. I suggest expanding on the definition and description of CAD. Authors define the components of metabolic syndrome and give a good description of its origin of definition. The statement on CAD only refers to “…enhance the heart muscles”.  What specifically does this mean – myocyte hypertrophy?  What is the definition of CAD and how is it determined clinically?  Are there degrees of CAD – mild, moderate, and severe?  Does the severity of MetS (e.g., number of MetS components present) correlate with the severity of CAD?   
  2. Thank you to the reviewer for these valuable comments. More information about CAD has been provided on pages 4 and 5, lines 104-113. The final question has been updated in key findings of the discussion section on page 9, lines 204-206.

  1. In the Results, authors state there are four studies from South Korea and three from the U.S. Where do the other three come from?
  2. Other each study from Malaysia, Netherland, and China and has been shown in Table 2.

  1. I agree that further studies are needed to understand the full significance of the components on MetS on CAD and other pathologies. Is it possible to discriminate among the components of MetS on CAD from the data presented in this study? Although the purpose of Funnel plots is for publication bias, can one interpret the asymmetry of high FBG in Figure 6 to be suggestive that this component might pose a higher risk for CAD than the high TG or high BP?
  2. The Funnel plots' purpose is for publication bias. However, the components of MetS on CAD’s risk discriminate that the asymmetry of high FBG and low HDL-C suggests that these components might pose a higher risk for CAD than the high TG high BP (see page 12, lines 281-283).

  1. Do the funnel plots of individual components and 2, 3, or 4 components not require diagonal lines?
  2. First paragraph page 13: “These evidence.” - This evidence
  3. It has been updated in the revised manuscript on page 11, line 251.

Thank you

Round 2

Reviewer 2 Report

authors replies to all the query raised and paper significantly improves and can now be accepted in its present form.